# A Revisit of Total Correlation in Disentangled Variational Auto-Encoder with Partial Disentanglement

## Abstract

A fully disentangled variational auto-encoder (VAE) aims to identify disentangled latent components from observations unsupervisedly. However, enforcing full independence between all latent components may be too strict for certain datasets. In some cases, multiple factors may be entangled together in a non-separable manner, or a single independent semantic meaning could be represented by multiple latent components within a higher-dimensional manifold. To address such scenarios with greater flexibility, we develop the Partially Disentangled VAE (PDisVAE), which generalizes the total correlation (TC) term in fully disentangled VAEs to a partial correlation (PC) term. This framework can handle group-wise independence and can naturally reduce to either the standard VAE or the fully disentangled VAE. Validation through three synthetic experiments demonstrates the correctness and practicality of PDisVAE. When applied to real-world datasets, PDisVAE discovers valuable information that is difficult to uncover with fully disentangled VAEs, implying its versatility and effectiveness.

## 1 Introduction

Disentangling independent latent components from observations is a desirable goal in representational learning (Bengio et al., 2013; Alemi et al., 2016; Schmidhuber, 1992; Achille & Soatto, 2017), with numerous applications in fields such as computer vision and image processing (Lake et al., 2017), signal analysis (Hyvärinen & Oja, 2000; Hyvarinen & Morioka, 2017), and neuroscience (Zhou & Wei, 2020; Yang et al., 2021; Wang et al., 2024; Calhoun et al., 2009). To disentangle latent components in an unsupervised manner, most models employ techniques that combine optimizing a variational auto-encoder (VAE) (Kingma, 2013) with an additional penalty term known as total correlation (mutual information) (Kraskov et al., 2004), classified as fully disentangled VAEs (Higgins et al., 2017; Kim & Mnih, 2018; Chen et al., 2018).

However, enforcing full independence among all latent components can be an overly strong assumption for certain datasets. For instance, consider the location coordinates $(x, y)$ of a set of points in a 2D plane. If the points are uniformly distributed within a square $[-1, 1] \times [-1, 1]$, the location distribution can be expressed as $p(x, y) = p(x)p(y)$, indicating that $x$ and $y$ are independent components. However, if the points are distributed in an irregular shape, such as a butterfly, the $(x, y)$ coordinates become entangled, resulting in $p(x, y) \neq p(x)p(y)$. In this case, the location information cannot be decomposed into two independent components but must be jointly represented by $(x, y)$ together. If the points also have attributes independent of their location, such as RGB color represented by a 3D vector, we then encounter the **group-wise independence**, where a rank-2 entangled group (location) is independent of a rank-3 entangled group (color).

Table 1: Comparison of methods. More details regarding these related methods are in Appendix A.1.

|  | full disentanglement | partial disentanglement |
|---|---|---|
| By prior (not flexible) | ICA | ISA-VAE |
| By extra penalty (flexible) | FactorVAE, $\beta$-TCVAE | Our PDisVAE |
| Others | citations and explanations listed in Appendix A.1 | |

To deal with such group-wise independence, one might consider a straightforward approach of using a fully disentangled method such as prior-based ICA (Hyvärinen & Oja, 2000) or penalty-based FactorVAE and $\beta$-TCVAE (Kim & Mnih, 2018; Chen et al., 2018) to impose marginal independence on between-group components (see Tab. 1). However, this is an insufficient condition for group-wise independence (see Sec. 3.1 and Appendix A.2 for details). Other approaches (row "others" in Tab. 1) either include semi-supervised learning to align the latent with the ground truth labels (e.g., Ahuja et al. (2022)) or do not exclusively penalize the term that is specifically for promoting independence. For example, $\beta$-VAE (Higgins et al., 2017; Burgess et al., 2018) penalizes the entire reverse KL term of the VAE target function, which is significantly less effective than FactorVAE and $\beta$-TCVAE that directly add a penalization, the total correlation (TC), for independence (Dubois et al., 2019). Hierarchical factorized VAE (Esmaeili et al., 2019) penalizes between-block latent independence, within-block latent independence, and their KL divergences w.r.t. their corresponding factorized priors. None of these methods directly deals with group-wise independence, where latent components within a group may be highly entangled. Among all these methods, ISA-VAE (Stühmer et al., 2020) is the first work that uses group-wise independent prior to achieve independence between latent groups, which can be viewed as an extension of nonlinear ICA, from full disentanglement to partial disentanglement. However, a predefined group-wise independent prior is sometimes inflexible to encompass various complicated latent distributions. Moreover, none of these methods rigorously validates or analyzes their effectiveness on a partially disentangled synthetic dataset.

To address these, we develop the **Partially Disentangled VAE (PDisVAE)**.

• First, it achieves group-wise independence by generalizing the total correlation (TC) penalty term in the target function of fully disentangled VAEs (Kim & Mnih, 2018; Chen et al., 2018) to partial correlation (PC), instead of using a rigidly defined group-wise independent prior used in ISA-VAE (Stühmer et al., 2020). PC explicitly penalizes group-wise independence while permitting within-group entanglement flexibly. This unified formulation of PC is flexible, and it encompasses both the standard VAE and fully disentangled VAEs.

• Second, we revisit the batch approximation method used for computing PC and TC from Chen et al. (2018) and Esmaeili et al. (2019). We theoretically prove that the importance sampling (IS) batch approximation from Esmaeili et al. (2019) is the optimal that is unbiased and has the lowest variance.

• Third, we are the first to conduct thorough experiments with proper metrics on three well-designed synthetic datasets with truth labels that are truly partially disentangled into groups. In particular, we create our **pdSprites** dataset, an extension of dSprites specifically designed to exhibit partially group-disentangled ground truth labels. Validation and analysis of these datasets demonstrate the superiority of our proposed PDisVAE in capturing group-wise independent latent factors.

In the following, we first introduce the background of fully disentangled VAEs. Then, we develop our PDisVAE and detail its techniques and properties. Lastly, we run experiments on three synthetic datasets and two real-world datasets to show that PDisVAE is effective in partially disentangling the latent components by groups.

## 2 BACKGROUNDS: FULLY DISENTANGLED VAEs

### 2.1 BY TOTAL CORRELATION (TC)

Given a dataset of observations $\left\{\boldsymbol{x}^{(n)}\right\}_{n=1}^{N}$ consisting of $N$ samples, fully disentangled VAEs identify $K$ statistically independent (disentangled) latent components, $z_1 \perp \cdots \perp z_K$, within the latent variable $\boldsymbol{z} \in \mathbb{R}^K$ that generate the observation $\boldsymbol{x} \in \mathbb{R}^D$, by optimizing

$$\mathcal{L} = \frac{1}{N} \sum_{n=1}^{N} \text{ELBO}\left(\boldsymbol{x}^{(n)}\right) - \beta \cdot \text{KL}\left(q(\boldsymbol{z}) \middle\| \prod_{k=1}^{K} q(z_k)\right), \tag{1}$$

where $\text{ELBO}(\boldsymbol{x}^{(n)}) = \mathbb{E}_{q(\boldsymbol{z}|\boldsymbol{x}^{(n)})}\left[\ln p\left(\boldsymbol{x}^{(n)}|\boldsymbol{z}\right)\right] - \text{KL}\left(q\left(\boldsymbol{z}|\boldsymbol{x}^{(n)}\right)\|p(\boldsymbol{z})\right)$ (Blei et al., 2017) is the standard VAE loss. In these formulae, $p(\boldsymbol{x}|\boldsymbol{z};\theta)$ is a decoder : $\mathbb{R}^K \to \mathbb{R}^D$ and $q(\boldsymbol{z}|\boldsymbol{x};\phi)$ is an encoder : $\mathbb{R}^D \to \mathbb{R}^K$. In Eq. (1) and the following, we omit $\theta$ in $p$ and $\phi$ in $q$ for simplification. The prior $p(\boldsymbol{z})$ is often chosen to be a standard normal prior. The second term in Eq. (1) is the total correlation (TC), where $q(\boldsymbol{z}) = \sum_{n=1}^{N} q\left(\boldsymbol{z}|\boldsymbol{x}^{(n)}\right) q\left(\boldsymbol{x}^{(n)}\right)$ is the aggregated posterior, followed by Makhzani et al. (2015). Since all data points are equally contributed, $q\left(\boldsymbol{x}^{(n)}\right) = \frac{1}{N}$, and hence $q(\boldsymbol{z})$ can be viewed as a Gaussian kernel density estimation from $\left\{\boldsymbol{z}^{(n)}\right\}_{n=1}^{N}$ in latent space. The TC term is designed to achieve the full latent disentanglement $q(\boldsymbol{z}) = \prod_{k=1}^{K} q(z_k) \iff z_1 \perp \cdots \perp z_K$.

## 2.2 BY A NON-GAUSSIAN PRIOR (ICA)

Another approach to achieving full disentanglement is the independent component analysis (ICA). Its core idea is "non-Gaussian is independent" (Hyvärinen & Oja, 2000; Hyvärinen et al., 2009). Therefore, the normally used standard Gaussian prior, although it can be factored into a product of marginals, does not enforce any independence. Hence, ICA replaces the standard normal prior with (a commonly used) logcosh prior: $p(\boldsymbol{z}) = \prod_{k=1}^{K} p(z_k) = \prod_{k=1}^{K} \frac{\pi \left( \operatorname{sech} \frac{\pi z_k}{2\sqrt{3}} \right)^2}{4\sqrt{3}}$. In traditional linear ICA, $\boldsymbol{x} = \boldsymbol{f}(\boldsymbol{z})$ where $\boldsymbol{f} : \mathbb{R}^K \to \mathbb{R}^D$ is a full-rank ($D = K$) linear deterministic mapping, and $p(\boldsymbol{x}|\boldsymbol{z}; \boldsymbol{f}) = \delta(\boldsymbol{x} - \boldsymbol{f}(\boldsymbol{z}))$ ($\delta$ is the Dirac delta function), then we can use maximum likelihood estimate (MLE) to learn $\boldsymbol{f}$ via the "change of variable" formula, $p(\boldsymbol{x}) = \int p(\boldsymbol{x}|\boldsymbol{z}; \boldsymbol{f}) p(\boldsymbol{z}) \, \mathrm{d}\boldsymbol{z} = \left| \det \frac{\mathrm{d}\boldsymbol{f}^{-1}}{\mathrm{d}\boldsymbol{z}} \right| \cdot p(\boldsymbol{f}^{-1}(\boldsymbol{x}))$, and recover $\boldsymbol{z} = \boldsymbol{f}^{-1}(\boldsymbol{x})$. For non-invertible non-linear $\boldsymbol{f}$, we can use a VAE with such a logcosh prior $p(\boldsymbol{z})$. We recognize this logcosh-priored VAE as the nonlinear ICA.

# 3 PARTIALLY DISENTANGLED VAE (PDISVAE)

## 3.1 PROBLEM DEFINITION

Although several approaches have been introduced in Sec. 2, a common issue among them is that they are all trying to find "fully disentangled (independent)" latent space. However, if the true latent variables are partially disentangled by groups, applying a fully disentangled method is hard to successfully recover the underlying latent structure accurately.

We first formally define partial disentanglement. Still, assume latent $\boldsymbol{z} \in \mathbb{R}^K$, but now the latent dimensions are disentangled by $G$ groups, while each group $g$ has its internal within-group rank $H_g$, satisfying $K = H_1 + \cdots + H_G$. For simplicity, we denote the $g$-th group as $\boldsymbol{z}_g = (z_{g,1}, \ldots, z_{g,H_g}) \in \mathbb{R}^{H_g}$, so that $\boldsymbol{z} = (\boldsymbol{z}_1, \ldots, \boldsymbol{z}_G)$. Then, the **partially disentangled** latent can be formulated as

$$\perp_{g=1}^{G} \boldsymbol{z}_g \iff p(\boldsymbol{z}) = \prod_{g=1}^{G} p(\boldsymbol{z}_g), \quad p(\boldsymbol{z}_g) \neq p(z_{g,1}) \cdots p(z_{g,H_g}), \quad \forall g \in \{1, \ldots, G\}. \quad (2)$$

This equation expresses that latent groups are independent of each other, but within each group, latent components may exhibit dependencies and may not be further disentangled. We refer to this as **group-wise independence** and present an example in Fig. 1.

To identify partially disentangled component groups, one might consider a straightforward approach: using existing methods to impose marginal independence on inter-group components. For instance, if we have $(z_1, z_2) \perp z_3$, one might attempt to apply existing algorithms to require $z_1 \perp z_3$ and $z_2 \perp z_3$. However, this is generally NOT correct since the former is a sufficient but not necessary condition ($\implies$) for the latter. A simple counterexample is $p(z_1, z_2, z_3)$ with $p(0,0,1) = p(0,1,0) = p(1,0,0) = p(1,1,1) = 0.25$. It can be verified that $(z_1, z_2) \not\perp z_3$, while $z_1 \perp z_3$ and $z_2 \perp z_3$. More detailed explanations are in Appendix A.2. Therefore, we must explicitly enforce $(z_1, z_2) \perp z_3$.

To explicitly require group-wise independence, there are still two ways—by a group-wise independent prior or by an extra penalty term to the loss function (see Tab. 1). Stühmer et al. (2020) developed ISA-VAE, extending from ICA, that utilizes the $L^p$-nested distribution (Fernández et al., 1995; Sinz & Bethge, 2010) as a group-wise independent prior to achieve the partial disentanglement. However, this approach still needs further experimental investigation (as it was not conducted in the ISA-VAE paper). Moreover, relying on a predefined prior to achieve group-wise independence might be overly rigid in some cases, similar to the logcosh prior in fully disentangled nonlinear ICA.

## 3.2 PARTIAL CORRELATION (PC)

Instead of using a prior, we develop the **Partially Disentangled VAE (PDisVAE)** that achieves the group-wise independence by an extra penalty term to the loss. Its target function

$$\mathcal{L} = \frac{1}{N} \sum_{n=1}^{N} \mathrm{ELBO}\left(\boldsymbol{x}^{(n)}\right) - \beta \cdot \mathrm{KL}\left(q(\boldsymbol{z}) \middle\| \prod_{g=1}^{G} q(\boldsymbol{z}_g)\right) \quad (3)$$

replaces the TC term in Eq. (1) with a partial correlation (PC) term. PC is responsible for disentangling independent groups. When $q(\boldsymbol{z}) = \prod_{g=1}^{G} q(\boldsymbol{z}_g)$, PC $= \mathrm{KL}\left(q(\boldsymbol{z}) \middle\| \prod_{g=1}^{G} q(\boldsymbol{z}_g)\right) = 0$. Otherwise, PC $> 0$ and is penalized by the hyperparameter $\beta > 0$.

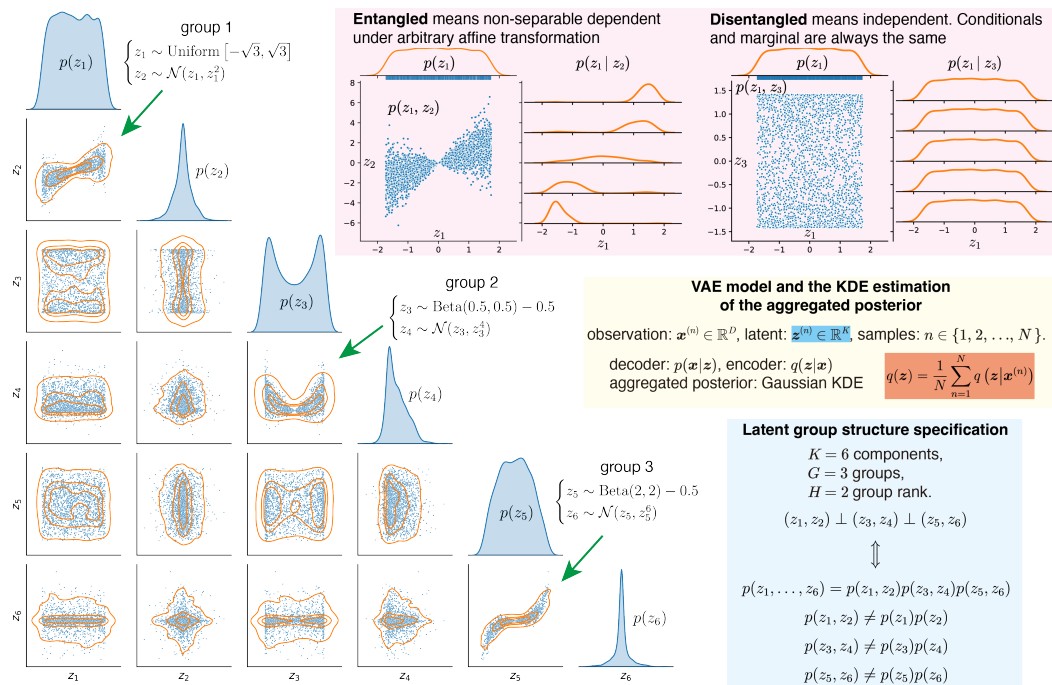

Figure 1: A synthetic dataset showing group-wise independent latent $(z_1, z_2) \perp (z_3, z_4) \perp (z_5, z_6)$, but within-groups are highly entangled. Marginal distributions are on the diagonal, and other off-diagonal positions represent the relationship between two latent components.

It is worth noting that when $G = 1$, PC $\equiv 0$ and Eq. (3) becomes the standard VAE objective function; when $G = K$, PC is just the total correlation (TC) and Eq. (3) becomes Eq. (1), the fully disentangled VAE loss. Compared with ISA-VAE (Stühmer et al., 2020), which relies on a predefined group-wise independent prior, utilizing PC to achieve group-wise independence offers greater flexibility by allowing the within-group disentanglement rank to vary, rather than being fixed to a specific rank $H$ in ISA-VAE. Specifically, when we don't know the true rank $H_{\text{true}}$ for a group, we can set a large enough group rank $H$ in the PDisVAE, and it will automatically detect the true effective group rank with the remaining $H - H_{\text{true}}$ dimensions as dummy variables. This flexibility and effectiveness will be validated through experiments.

### 3.3 BATCH APPROXIMATION

During training, strictly computing the aggregated marginal/group posterior of the form $q(z) = \sum_{n=1}^{N} q(z|x^{(n)})q(x^{(n)}) = \frac{1}{N} \sum_{n=1}^{N} q(z|x^{(n)})$ might be unfeasible, since we only have a batch, denoted as $\mathcal{B}_M := \{n_1, n_2, \ldots, n_M\}$ without replacement. Although Chen et al. (2018) proposed minibatch weighted sampling (MWS) and minibatch stratified sampling (MSS), we argue that the **importance sampling (IS)** (first proposed by Esmaeili et al. (2019)) is theoretically more effective.

Specifically, when we only have a batch $\mathcal{B}_M \subsetneq \{1, \ldots, N\}$ and a sampled $z \sim q(z|n_*)$, where $n_*$ is a specific example point in $\mathcal{B}_M$, $q(z|n_*)$ is more likely to be greater than $q(z|n \neq n_*)$ since $z$ is sampled from $q(z|n_*)$. Therefore, we want the remaining $M - 1$ points in $\mathcal{B}_M \setminus \{n_*\}$ to represent the entire dataset excluding $n_*$, i.e., $\{1, 2, \ldots, N\} \setminus \{n_*\}$. Hence, an approximation of $q(z)$ at $z \sim q(z|n_*)$ could be

$$\hat{q}(z) = \frac{1}{N} q(z|n_*) + \sum_{n \in (\mathcal{B}_M \setminus \{n_*\})} \frac{N-1}{M-1} \frac{1}{N} q(z|n). \quad (4)$$

Notably, IS is theoretically more stable than MSS due to the following theorem.

**Theorem 3.1.** *The effectiveness of the IS estimator is higher than that of the MSS estimator, measured by the variance of the estimator, satisfies* $\mathrm{Var}[\mathrm{IS}] < \mathrm{Var}[\mathrm{MSS}], \forall M > 2$.

Appendix A.3 includes the complete derivation, the proof of its optimality, and an empirical evaluation of the three estimators, which constitute one of the core contributions of this work.

## 4 EXPERIMENTS

**Methods for comparison.**
• Standard **VAE** (Kingma, 2013): Theoretically, standard VAE does not have disentanglement ability.
• **ICA**: The logcosh-priored VAE for doing non-linear ICA inspired by Hyvärinen & Oja (2000).
• **ISA-VAE** (Stühmer et al., 2020): This is the VAE that using the $L^p$-nested prior to achieve group-wise independence.
• $\beta$**-TCVAE** (Chen et al., 2018): This method penalizes an extra TC term to achieve full disentanglement. It is theoretically equivalent to FactorVAE (Kim & Mnih, 2018).
• **PDisVAE**: Our method penalizes the PC term to achieve partial disentanglement, providing a flexible approach to group-wise independent latent. It reduces to the standard VAE when the number of groups $G = 1$; and reduces to the fully disentangled VAE when $G = K$ (i.e., the number of groups equals the latent dimensionality). Additionally, it inherently supports within-group rank deficiency.

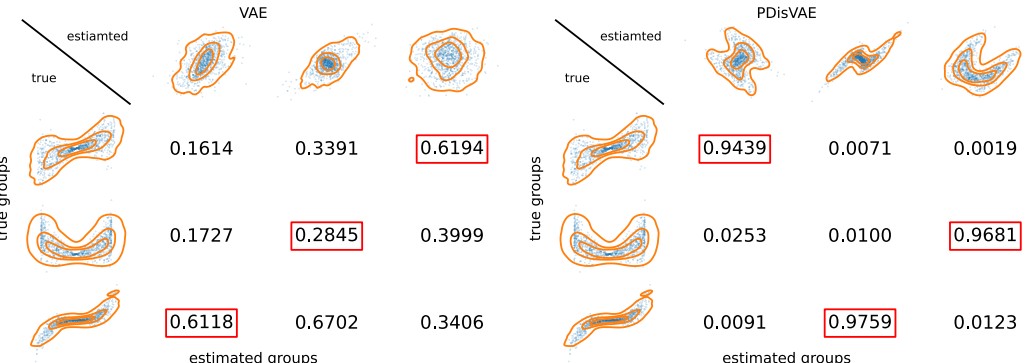

Figure 2: The latent alignment procedure illustrated by the R2 alignment matrix. The best match is marked by the red squared linear least-squares $R^2$ score.

### 4.1 SYNTHETIC VALIDATION: GROUP-WISE INDEPENDENT

**Dataset.** To validate that only PDisVAE is capable of dealing with group-wise independent datasets, we use our created dataset in Fig. 1 consisting of $N = 2000$ points in $K = 6$ latent space $\boldsymbol{z}^{(n)} \in \mathbb{R}^6$, where three groups are independent of each other $(z_1, z_2) \perp (z_3, z_4) \perp (z_5, z_6)$, but components within each group are highly entangled. The observations $\boldsymbol{x}$ are linearly mapped from the latents $\boldsymbol{z}$ to a $D = 20$ dimensional space $\boldsymbol{x}^{(n)} \in \mathbb{R}^{20}$, and then Gaussian noise $\epsilon_d^{(n)} \overset{i.i.d.}{\sim} \mathcal{N}\left(0, 0.5^2\right)$ is added.

**Experimental setup.** For each method, we use Adam (Kingma, 2014) to train a linear encoder and a linear decoder (since the true generative process is linear) for 5,000 epochs. The learning rate is $5 \times 10^{-4}$ and the batch size is 128. For $\beta$-TCVAE and PDisVAE, the TC/PC penalty is set as $\beta = 4$. This is supported by Dubois et al. (2019), the $\beta$ selection in $\beta$-TCVAE (Chen et al., 2018), and our cross-validation result (Fig. 7 in Appendix A.4.1) in the ablation study. Each method is run 10 times with different random seeds.

**Partial disentanglement evaluation.** When there is no ground truth latent groups, we can use the **PC on the test set** as a metric to evaluate whether the latent space has group-wise independent structure. When the ground truth exists, we can match the estimated latent groups $\{\boldsymbol{z}_1^{(n)}\}_{n=1}^N, \ldots, \{\boldsymbol{z}_G^{(n)}\}_{n=1}^N$ to the true groups $\{\boldsymbol{z}_1'^{(n)}\}_{n=1}^N, \ldots, \{\boldsymbol{z}_G'^{(n)}\}_{n=1}^N$ correspondingly. Examples of this aligning procedure are illustrated in Fig. 2. Specifically, we form an $\mathbf{R2} \in (-\infty, 1]^{G \times G}$ matrix whose entry $(g_1, g_2)$ is the $R^2$ score by aligning the estimated latent group $\boldsymbol{z}_{g_2}^{(n)}$ to the true $\boldsymbol{z}_{g_1}'^{(n)}$ via a linear least-squares fit. We then solve a linear-sum assignment problem (Crouse, 2016) to find a one-to-one correspondence matching between true groups $g'$ and estimated groups $g$ that maximizes the total $R^2$, and report the mean $R^2$ over these matched pairs.

For VAE, there is no disentanglement assumption, and hence the estimated latent does not contain any type of disentangled structure. For ICA and $\beta$-TCVAE, they assume dimension-wise independence rather than the desired group-wise independence. None of these methods has a group-wise structure and hence cannot provide estimated latent groups theoretically. Therefore, we grid search all possible groupings and pick the one that has the best alignment in a post-hoc way. In this experiment, we need

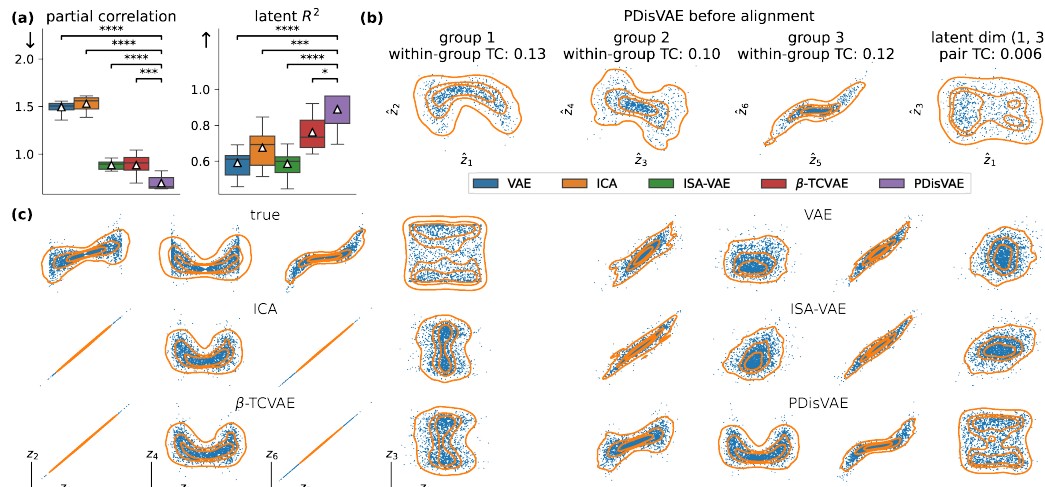

Figure 3: **(a)**: The PC of the estimated latent and the latent $R^2$ after alignment to the true latent (Fig.1, with pair-wise $t$-test showing the significance level. **(b)**: The estimated latent of PDisVAE before aligning to the true latent. In each pair, the TC shows the minimum TC under all possible linear transformations. **(c)**: Estimated latent after aligning to the true latent for various methods. Left three columns: the three independent groups; right one column: a between-group component pair.

to assign $K = 6$ estimated latent dimensions into $G = 3$ groups, and each of them contains $H = 2$ dimensions (with $K = G \times H$). Therefore, the number of all possible grouping combinations is $\frac{\prod_{g=1}^{G} \binom{(G+1-g)H}{H}}{G!} = \frac{K!}{G!(H!)^G} = 15$ for this example. Such a high complexity explicitly demonstrates the theoretical defect of methods without a flexible or proper group-wise independence assumption. Compared with them, PDisVAE completely eliminates the need for post-hoc analyses.

**Results.** The PC box plot in Fig. 3(a) shows that PDisVAE achieves the lowest PC, implying that PDisVAE disentangles latent in groups the best, while others do not provide proper group-wise structures in latent space. Since the PC on the test set is also evaluated numerically, we compute the PC of the true latent as a sanity check, obtaining $0.332 \pm 0.006$. The magnitude of this value confirms that the PC achieved by PDisVAE indeed reflects a substantially better latent group structure compared to other methods. This magnitude.

The VAE reconstruction $R^2$ between the true and the VAE reconstructed observation of all methods is approximately 0.97, indicating that all methods can reconstruct the observation perfectly. However, their learned latent representations are different. Since this is a synthetic dataset and a model match experiment, we can align the estimated latent groups to their corresponding true latent groups to further validate the correctness of the latent estimation. The alignment procedure visualized in Fig. 2 indicates that no matter how we partition the estimated latent dimensions into three groups, each estimated group contains some information from all three true groups. However, each estimated latent group from PDisVAE exclusively contains nearly complete information from one particular true group, which forms the corresponding alignment result.

The latent $R^2$ boxplot in Fig. 3(a) and latent plots in Fig. 3(c) summarize that PDisVAE recovers the latent more accurately than others. Among the alternatives, $\beta$-TCVAE is better than ISA-VAE, ICA, and VAE. Although ISA-VAE is designed to find group-wise independent latent, its performance is not ideal when facing data generated from group-wise independent ground truth latent in practice, due to the predefined group-wise independent prior in ISA-VAE differing substantially from the true underlying latent groups. In contrast, PDisVAE does not impose a rigid prior, allowing greater flexibility to accommodate diverse latent structures. Another important message from these results reminds us that, even if a fully disentangled method such as ICA and $\beta$-TCVAE finishes, it cannot provide a reliable latent structure if the fully disentangled assumption itself is wrong.

An immediate question that arises is, how to check within-group latent estimated by PDisVAE is truly highly entangled and cannot be further decomposed, especially when there is no true latent. The minimum within-group TCs in Fig. 3(b) are all significantly greater than zero, indicating highly

entangled groups that cannot be further decomposed. In contrast, the near-zero pairwise TCs between groups suggest independence across groups.

**Flexibly reduce to the fully independent case.** To validate that PDisVAE can flexibly get the same results as from a fully disentangled VAE when the latent is fully independent, we create a dataset that is generated from fully independent latent (Fig. 8(a) and Fig. 9) and apply different methods to it. The PC box plot and latent $R^2$ plot in Fig. 8(b) show that both $\beta$-TCVAE and PDisVAE achieve the lowest partial correlation and the highest latent $R^2$ on this fully disentangled dataset, which implies that PDisVAE automatically reduces to a fully independent result if the group rank is deficient. In general, the actual group rank can be detected by PDisVAE and if the true group rank is less than the specified group dimensionality, dummy estimated latents will be complemented in the corresponding group. More details are in Appendix A.4.2.

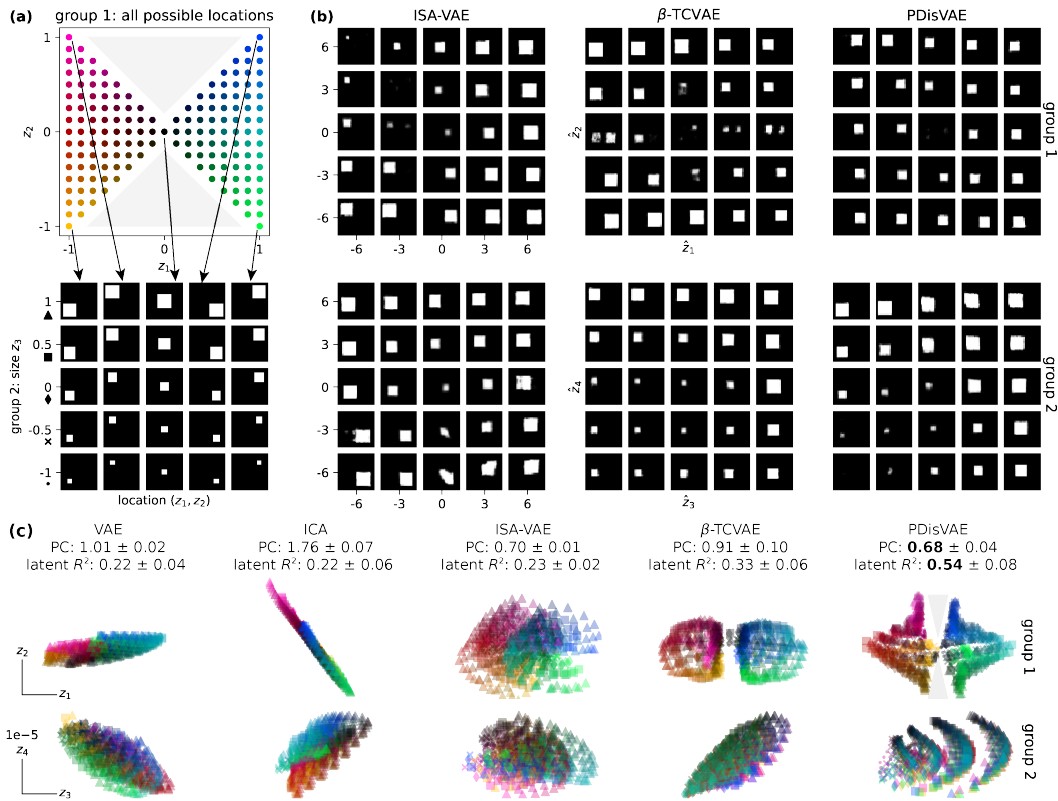

Figure 4: **(a)**: Latent and observation generating process. Locations $(z_1, z_2)$ are entangled and uniformly distributed in a restricted region. Color encodes location, with the upper and lower gray triangular areas being empty. The size $z_3$ is evenly distributed across five scales, represented by different markers, and is independent of the location. **(b)**: The reconstructed images by varying one of the latent groups (($\hat{z}_1, \hat{z}_2$) or ($\hat{z}_3, \hat{z}_4$)). **(c)**: The latent plot and their corresponding PC and latent $R^2$. The color-location and marker-size correspondences are identical to (a).

### 4.2 SYNTHETIC APPLICATION: PARTIAL DSPRITES

**Dataset.** To understand the application scenario of PDisVAE, we created a synthetic dataset called partial dsprites (pdsprites), inspired by Matthey et al. (2017). Unlike the original dsprites, which features six fully independent latent dimensions, we only keep three latent components: $x$-location ($z_1$), $y$-location ($z_2$), and size ($z_3$), where $x$ and $y$ locations are entangled (not independent) with each other while this group is independent to the size, i.e., $(z_1, z_2) \perp z_3$. The generating process is depicted in Fig. 4(a), resulting in 805 gray-scaled images of shape $32 \times 32$.

**Experimental setup.** For each method, we use Adam to train a deep CNN VAE (Burgess et al., 2018) for 5,000 epochs with a learning rate of $1 \times 10^{-3}$. For $\beta$-TCVAE and PDisVAE, the TC/PC coefficient is set as $\beta = 4$. Given the true latent is $(z_1, z_2) \perp z_3$, learning two rank-2 groups

$(K = 4 = G \times H = 2 \times 2)$ should be able to find one group representing the location of the square and another rank-deficient group (contains a dummy latent component) representing the size of the square. Note that this setup is a model mismatch case, as we do not know the exact observation generating function $\boldsymbol{f}$; we only know the semantic relationship between $\boldsymbol{z}$ and $\boldsymbol{x}$.

**Results.** Fig. 4(c) shows the estimated latent from all methods after alignment. PDisVAE has the highest latent $R^2$ and the second lowest PC. Notably, PDisVAE successfully discovers two empty areas in the upper and lower gray triangular regions in group 1, reflecting the true latent distribution depicted in Fig. 4(a). Additionally, although we specify two rank-2 groups, PDisVAE automatically finds the group for "size" contains one effective component that reflects the "size" and one dummy component. Specifically, it captures leveled size scales in $z_3$, showing smaller sizes for smaller $z_3$ and larger sizes for larger $z_3$, making it the closest representation of the true $z_3$ compared to other methods. This further demonstrates the flexibility of PDisVAE in scenarios where the true group specifications are unknown, as in real-world datasets. By setting a sufficiently large group rank for each group, PDisVAE can automatically infer the effective rank within each group. Appendix A.4.3 contains more plots and quantitative comparisons.

Fig. 4(b) shows the reconstructed images by varying each of the two groups found by $\beta$-BTCVAE and PDisVAE, respectively. Group 1 from PDisVAE represents the location, with an empty center due to fewer observation samples in that area (see the region around $(z_1, z_2) = (0, 0)$ in Fig. 4(a)). Besides, the square is expected not to appear in the top middle or bottom middle of the image, since no observation in the dataset appears in those regions. The size is embedded in group 2, roughly along the $\hat{z}_4$ direction. In contrast, $\beta$-TCVAE mixes size and location in both groups because it enforces independence across all four components, which is incompatible with the fact that two location components are entangled together and independent of the third size component.

### 4.3 REAL-WORLD APPLICATIONS

We evaluate the performance and flexibility of PDisVAE on two real-world applications. For real-world datasets, the true latent structure is unknown. While PDisVAE can theoretically tolerate over-specified group ranks, excessively large settings degrade training efficiency and model quality. To systematically examine the impact of group specification, we fix the total latent dimensionality $K$ and vary $(G, H)$. This design allows us to study how different group assumptions affect performance while enabling fair comparisons with the standard VAE ($G = 1$) and the fully disentangled VAE ($G = K$) under the same latent capacity $K$.

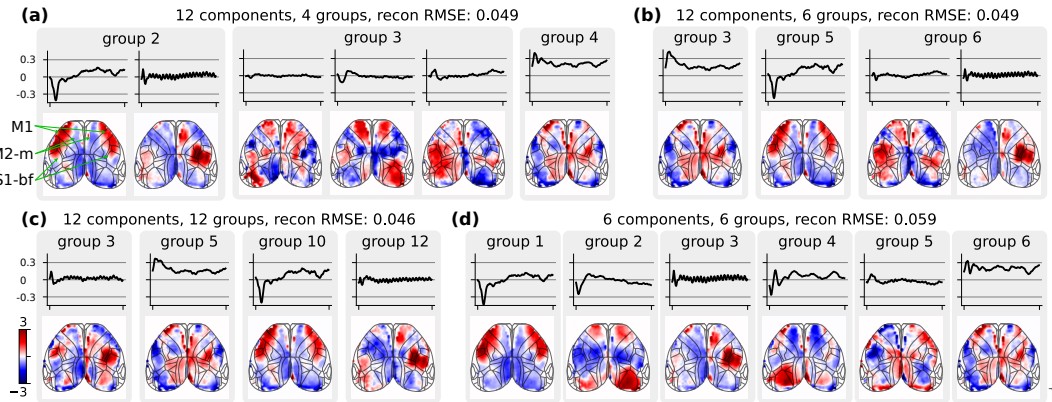

Figure 5: Brain maps $\left\{ \boldsymbol{z}_g^n \right\}_{n=1}^{50 \times 50}$ and the corresponding time series $\boldsymbol{A}_{:,g}$ from the learned groups by different PDisVAE configurations $(K, G)$, i.e., $K$ components, $G$ groups, and the group rank is $H = K/G$. Some groups contain dummy dimensions, so the effective group rank is lower than the specified group rank, and hence we only show those effective components.

**Mouse dorsal cortex voltage imaging.** The dataset used in this study is a trial-averaged voltage imaging (method by Lu et al. (2023)) sequence from a mouse collected by us. It comprises 150 frames of $50 \times 50$ dorsal cortex voltage images, recorded while the mouse was subjected to a left-side air puff stimulus lasting 0.75 seconds. Each pixel is treated as a sample, and a linear model $\boldsymbol{x} \sim \mathcal{N}(\boldsymbol{Az}, \sigma^2 \boldsymbol{I})$ is learned. We investigate different numbers of groups $G \in \{1, 2, 3, 4, 6, 12\}$ while keeping the

number of components constant at $K = 12$. Additionally, we explore fully disentangled models by varying $K \in \{1, 2, 3, 4, 6, 12\}$ with $G = K$. The training procedures are similar to the previous experiments (see code for details).

Figure 5 shows the brain maps and corresponding time series learned from various PDisVAE configurations $(K, G)$. Learning $K = 12$ components with different $G$ groups (Fig. 5(a,b,c)) yields similar reconstruction RMSEs ($\approx 0.047$), but results in different latent representations. Assuming $G = 12$ as a fully disentangled model (Fig. 5(c)) is overly restrictive, as both group 3 and group 12 contain oscillations in the right primary somatosensory cortex-barrel field (S1-bf) and secondary motor cortex-medial (M2-m), demonstrating a lack of independence between these components. This configuration implies that there are not 12 independent components within this neural data. Conversely, assuming $G = 4$ groups (Fig. 5(a)) is insufficient, as group 2 mixes not only the oscillatory signals right S1-bf and M2-m but also signals from other regions like the right primary motor cortex (M1). This implies a failure to capture the complete scope of independence in the data. A $G = 6$ grouping (Fig. 5(b)) presents a more balanced approach. This model consists of six independent groups, each expressed by two latent components. Specifically, group 3's S1-bf and M2-m remain active, indicating these areas are stimulated during the air puff; group 6 is primarily responsible for the oscillations in S1-bf and M2-m, with minimal interference from the M1 signal. Moreover, the brain maps in group 2 from the 4-group configuration are effectively delineated into groups 5 and 6 in the 6-group configuration, further affirming the relative independence of M1 from S1-bf and M2-m during stimulus exposure. The fully independent model with $(K, G) = (6, 6)$ (Fig. 5(d)) indicates that two components per group are necessary for accurate reconstruction. Specifically, having only one component per group is insufficient to reconstruct the raw video, as the RMSE for $(6, 6)$ is 0.059, which is significantly higher than the 0.049 RMSE for $(12, 6)$. The group reconstruction videos in the supplementary materials offer a more intuitive illustration of the full contribution of each group.

**CelebA.** The dataset contains 202,599 face images (Liu et al., 2015), cropped and rescaled to $(3, 64, 64)$. The encoder and decoder are deep CNN-based image-nets (Burgess et al., 2018). We fix the latent dimensionality $K = 12$ and vary the number of groups $G \in \{1, 2, 3, 4, 6, 12\}$. Training settings are similar to the previous experiments. Figures for this experiment are in Appendix A.4.

Fig. 11(a) shows the reconstructed images by varying each of the $K = 12$ components while fixing others as zero, for $G \in \{4, 6, 12\}$. The group meanings are annotated on the left. Particularly, with 4 or 6 groups, some attributes are represented by a group of higher rank rather than a single latent component, such as background color. Certain attributes are dependent on each other represented by a group, like the face color & hair color in the $G = 4$ setting. These important interpretations are harder to find by the fully disentangled $G = 12$ setting. Besides, a fully disentangled VAE may fail to ensure perfect independence if the component setting and the true latent factor are largely mismatched (which is also hard to determine), like gender 1 and gender 2 in the $G = 12$ setting.

To understand how one semantic attribute is represented by multiple components within a group, we use background color as an example. The $G = 12$ groups setting in Fig. 11(a) shows that the background color is represented by a single component, which restricts the expression to a 1D color manifold as shown in $G = 12$ HSV cylinder in Fig. 11(b), which is not reasonable. With multiple latent components in a group representing background color, the background color can be expressed in 2D or 3D color manifolds as shown in $G = 6$ and $G = 4$ HSV cylinders, offering a more expressive and realistic representation. Results from all group settings are displayed in Fig. 12 in Appendix A.4.

## 5 CONCLUSION

In this work, we develop PDisVAE, a flexible approach to modeling group-wise independence, which is often more realistic than full independence. PDisVAE generalizes to standard or fully disentangled VAEs by setting the number of groups to 1 or to the latent dimensionality, and it permits dummy components when learned latents are fewer than the specified group rank.

A potential limitation of PDisVAE is the need for an adequate number of groups and internal group rank to accurately express the disentangled latent space, especially when the data demands it, yet such guidance is often unavailable. While setting a large enough number is theoretically feasible, it hampers the training efficiency and the model quality in practice. Addressing this may require either trying different configurations or developing techniques for automatic group specification adjustment during training in future works. More discussions are in Appendix A.5.

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

# A APPENDIX

## A.1 RELATED WORKS

Realizing there are a lot of methods related to latent disentanglement, we go through the methods listed in Tab. 1, summarize their contributions and differences as follows.

- ICA (Hyvärinen & Oja, 2000): Traditional ICA uses a non-Gaussian prior to achieve full disentanglement since independence is non-Gaussian from the statistical perspective. However, the choice of the non-Gaussian prior is critical and might be too rigid, hurting the flexibility of the method.
- FactorVAE (Kim & Mnih, 2018) [3] $\beta$-TCVAE Chen et al. (2018): These two papers start from the statistical definition of full independence to add an extra total correlation to achieve full independence rigorously. The only difference between these two papers is their implementations of minimizing TC.
- ISA-VAE (Stühmer et al., 2020): ISA-VAE realized the commonly existing group-wise independence (partial disentanglement) in the real-world data. It utilizes a group-wise independent prior called $L^p$-nested distribution to achieve the partial disentanglement. However, they did not validate their approach on partially disentangled synthetic datasets, but merely evaluated their approach using fully disentangled assumptions for dSprites and CelebA datasets.
- $\beta$-VAE (Burgess et al., 2018): Directly penalize the KL divergence of the VAE ELBO loss, in which total correlation (TC) is implicitly penalized. This approach has been proven to be worse than $\beta$-VAE and FactorVAE.
- Locatello et al. (2019): This research presented common challenges in finding disentangled latent through an unsupervised approach, implying supervision with semantic latent labels might be necessary under the assumption of full latent disentanglement. This also gives us a hint that full disentanglement might be a strong and inappropriate assumption and could result in poor latent interpretation.
- Ahuja et al. (2022): This paper uses weak supervision from observations generated by sparse perturbations of the latent variables, which requires auxiliary information about the latent variables.
- $\alpha$-VAE (Meo et al., 2024): This paper replace the traditional TC term with a novel TC lower bound to achieve not only disentanglement but generalized observation diversity.
- Bhowal et al. (2024): This paper claims that VAE with orthogonal structure could also achieve latent full disentanglement.
- Hsu et al. (2024): The full disentanglement is achieved by a technique called latent quantization. The approach is quantizing the latent space into discrete code vectors with a separate learnable scalar codebook per dimension. Besides, weight decay is also applied to the model regularization for better full disentanglement.
- Hierarchical factorized VAE (Esmaeili et al., 2019): This paper has a structured decomposition of the ELBO target function, and penalizes different terms to achieve independence between blocked factors.
- HSIC (Lopez et al., 2018): This paper deals with independence between a group pair, rather than independence between all groups.

## A.2 MARGINAL INDEPENDENCE

This part explains the sufficient but not necessary relationship between "group-wise independence" and "marginal independence". Consider a latent variable $z \in \mathbb{R}^M$ that contains $M$ components that are independent between $G$ groups. The formal expression is

$$\underset{g=1}{\overset{G}{\perp}} \left( z_{g,1}, \ldots, z_{g,H_g} \right) \implies \bigwedge_{i \in g_1, j \in g_2, g_1 \neq g_2} z_i \perp z_j, \tag{5}$$

but not vice versa. We start from the simple counterexample mentioned in Sec. 3.1 to explain why group-wise independence is a sufficient but not necessary condition of marginal independence.

Consider three random variables $z_1, z_2, z_3$ that follow the joint distribution shown in Tab. 2. Notice that $z_3$ is actually the exclusive or of the two others, i.e., $z_3 = \text{XOR}(z_1, z_2)$. It is obvious that $z_3 \not\perp (z_1, z_2)$ since when $z_1$ and $z_2$ are different, $p(z_3|z_1, z_2)$ is a discrete Dirac delta function at $z_3 = 0$; but when $z_1$ and $z_2$ are the same, $p(z_3|z_1, z_2)$ is a discrete Dirac delta function at $z_3 = 1$. Marginally, however, $z_1 \perp z_3$ and $z_2 \perp z_3$, since $p(z_3|z_1)$ is always a $p = 0.5$ Bernoulli distribution regardless of the value of $z_1$. The same arguments are also applicable to $z_2 \perp z_3$. Therefore, this counterexample shows that $z_1 \perp z_3, z_2 \perp z_3 \implies\!\!\!\!\!/ \ (z_1, z_2) \perp z_3$. In other words, marginal independence does not imply group-wise independence.

Another way of checking this example is by the following theorem.

**Theorem A.1.** $(x_1, \ldots, x_I) \perp (y_1, \ldots, y_J) \iff \left( f(x_1, \ldots, x_I) \perp g(y_1, \ldots, y_J) \ \forall \text{ measurable} \right.$ *functions $f$ and $g$*$\left. \right)$.

*Proof.* The $\implies$ is obvious. To prove $\impliedby$, simply taking $f$ and $g$ to be identity function, i.e., $f(x_1, \ldots, x_I) = (x_1, \ldots, x_I), g(y_1, \ldots, y_J) = (y_1, \ldots, y_J)$. $\square$

To check the example, consider the distribution of $(z_1 + z_2)$. $p(z_3|(z_1 + z_2) = 0)$ is a discrete Dirac delta function at $z_3 = 1$, which is different from $p(z_3|(z_1 + z_2) = 1)$ is a discrete Dirac delta function at $z_3 = 0$. Therefore, $(z_1, z_2) \not\perp z_3$.

To rigorously diagnose where $\impliedby$ breaks, we can write

$$p(z_1, z_2, z_3) = p(z_1|z_2, z_3)p(z_2, z_3) = p(z_1|z_2, z_3)p(z_2)p(z_3). \tag{6}$$

Note that in the last term, $p(z_1|z_2, z_3) \neq p(z_1|z_2)$. Specifically, $z_3$ cannot be removed just because of $z_1 \perp z_3$.

Table 2: The distribution table of $p(z_1, z_2, z_3)$.

| $z_1$ | $z_2$ | $z_3$ | $p(z_1, z_2, z_3)$ |
|---|---|---|---|
| 0 | 0 | 1 | 0.25 |
| 0 | 1 | 0 | 0.25 |
| 1 | 0 | 0 | 0.25 |
| 1 | 1 | 1 | 0.25 |

## A.3 BATCH APPROXIMATION

Table 3: Comparison of three batch approximation approaches.

|  | mean | variance |
|---|---|---|
| MWS | biased | |
| MSS | unbiased | $\text{Var}[\text{MSS}] = \text{Var}[\text{IS}] + \frac{M-2}{M(M-1)}$ |
| **IS** | unbiased | $\text{Var}[\text{IS}] = \frac{(N-M)^2}{M^2(M-1)}$ |

### A.3.1 IMPORTANCE SAMPLING

Although Eq. (4) in the main text intuitively gives the batch approximation, we still need a rigorous derivation to prove that this is exactly the importance sampling (IS) we want. First, we have the aggregated posterior that can be expressed in different ways:

$$q(z) = \sum_{n=1}^{N} q(z,n) = \sum_{n=1}^{N} q(z|n)q(n) = \frac{1}{N} \sum_{n=1}^{N} q(z|n) = \mathbb{E}_{q(n)}[q(z|n)]. \tag{7}$$

However, to not confuse readers, we will keep the form $q(z) = \sum_{n=1}^{N} q(z,n)$ until the last step.

When we have a batch of size $M$: $\mathcal{B}_M := \{n_1, n_2, \ldots, n_M\}$ (without replacement) and a particular sampled $z \sim q(z|n_*)$, where $n_* \in \mathcal{B}_M$, we want the importance sampling approximation of $q(z)$. According to Monte Carlo estimation,

$$\hat{q}(z) = \frac{1}{M} \sum_{m=1}^{M} \frac{q(z,n_m)}{r(n_m)}, \tag{8}$$

where $r$ is the proposal distribution. Note that $r(n_m) \neq \frac{1}{N}$, $\forall n_m \in \mathcal{B}$, since we must have $n_* \in \mathcal{B}_M$. Therefore, we need to understand the distribution of $r(n_m)$.

First, since we must have $n_* \in \mathcal{B}_M$, and the Monte Carlo estimation is the average on $\mathcal{B}_M$,

$$r(n_*) = \underbrace{1}_{n_* \text{ must be in } \mathcal{B}_M} \times \underbrace{\frac{1}{|\mathcal{B}_M|}}_{n_* \text{ is a Monte Carlo sample from} \mathcal{B}_M} = \frac{1}{M}. \tag{9}$$

Second, for other $n_m \notin \mathcal{B}_M$,

$$r(n_m) = \underbrace{\frac{\binom{N-2}{M-2}}{\binom{N-1}{M-1}}}_{n_m \text{ is selected in batch } \mathcal{B}_M} \times \underbrace{\frac{1}{|\mathcal{B}_M|}}_{n_m \text{ is a Monte Carlo sample from} \mathcal{B}_M} = \frac{M-1}{N-1}\frac{1}{M}. \tag{10}$$

$\binom{N-1}{M-1} = \frac{(N-1)!}{(M-1)!((N-1)-(M-1))!}$ is the number of all possible combinations of $\mathcal{B}_M$ that already contains $n_*$ (so we choose $M-1$ from the remaining $N-1$). $\binom{N-2}{M-2} = \frac{(N-2)!}{(M-2)!((N-2)-(M-2))!}$ is the number of all possible combinations of $\mathcal{B}_M$ that already contains $n_*$ and also contains $n_m$ (so we choose $M-2$ from the remaining $N-2$). Finally, we have

$$\begin{aligned}
\hat{q}(z) &= \frac{1}{M} \sum_{m=1}^{M} \frac{q(z,n_m)}{r(n_m)} \\
&= \frac{1}{M}\frac{q(z|n_*)q(n_*)}{r(n_*)} + \sum_{n_m \in (\mathcal{B}_M \setminus \{n_*\})} \frac{1}{M}\frac{q(z|n_m)q(n_m)}{r(n_m)} \\
&= \frac{1}{M}\frac{q(z|n_*)\frac{1}{N}}{\frac{1}{M}} + \sum_{n_m \in (\mathcal{B}_M \setminus \{n_*\})} \frac{1}{M}\frac{q(z|n_m)\frac{1}{N}}{\frac{M-1}{N-1}\frac{1}{M}} \\
&= \frac{1}{N}q(z|n_*) + \sum_{n_m \in (\mathcal{B}_M \setminus \{n_*\})} \frac{N-1}{M-1}\frac{1}{N}q(z|n_m).
\end{aligned} \tag{11}$$

### A.3.2 VARIANCE

From Chen et al. (2018), without loss of generality, assume $n_* = n_1$ and

$$
\begin{aligned}
\text{MSS} =& \frac{1}{N} q(z|n_*) + \sum_{m=2}^{M-1} \frac{1}{M-1} q(z|n_m) + \frac{N-M+1}{N(M-1)} q(z|n_M) \\
=& \frac{1}{N} q(z|n_*) + \sum_{m=2}^{M-1} \frac{N}{M-1} \frac{1}{N} q(z|n_m) + \frac{N-M+1}{(M-1)} \frac{1}{N} q(z|n_M).
\end{aligned} \tag{12}
$$

A sketch to compute the variances of the two methods is to think of them as sampled datasets of size $M$. Specifically, for IS, the inverse importance weights are a dataset of $\text{IS}_0 :=$
$\left\{ 1, \underbrace{\frac{N-1}{M-1}, \ldots, \frac{N-1}{M-1}}_{M-1} \right\}$. For, MSS, the inverse importance weights are a dataset of $\text{MSS}_0 :=$
$\left\{ 1, \underbrace{\frac{N}{M-1}, \ldots, \frac{N}{M-1}}_{M-2}, \frac{N-M+1}{M-1} \right\}.$

There means are all $\frac{N}{M}$, since

$$
\begin{cases}
\overline{\text{MSS}_0} = \frac{1}{M} \left( 1 + (M-2)\frac{N}{M-1} + \frac{N-M+1}{M-1} \right) = \frac{N}{M} \\
\overline{\text{IS}_0} = \frac{1}{M} \left( 1 + (M-1)\frac{N-1}{M-1} \right) = \frac{N}{M}
\end{cases} \tag{13}
$$

Now we compute their variances.

$$
\begin{aligned}
\text{Var}[\text{MSS}] &\propto \text{Var}[\text{MSS}_0] \\
&= \frac{1}{M} \left[ \left( 1 - \frac{N}{M} \right)^2 + (M-2)\left( \frac{N}{M-1} - \frac{N}{M} \right)^2 + \left( \frac{N-M+1}{M-1} - \frac{N}{M} \right)^2 \right] \\
&= \frac{2M^2 - (2N+2)M + N^2}{M^2(M-1)}.
\end{aligned} \tag{14}
$$

$$
\begin{aligned}
\text{Var}[\text{IS}] &\propto \text{Var}[\text{IS}_0] \\
&= \frac{1}{M} \left[ \left( 1 - \frac{N}{M} \right)^2 + (M-1)\left( \frac{N-1}{M-1} - \frac{N}{M} \right)^2 \right] \\
&= \frac{(N-M)^2}{M^2(M-1)}.
\end{aligned} \tag{15}
$$

Since

$$
\text{Var}[\text{IS}_0] - \text{Var}[\text{MSS}_0] = \frac{2-M}{M(M-1)} \leqslant 0, \ \forall M \geqslant 2, \tag{16}
$$

the effectiveness of IS is higher, and hence IS is a more stable approximation than MSS.

### A.3.3 EMPIRICAL EVALUATION

To validate the aforementioned superiority of our IS batch estimation method, we simulate a dataset consisting of 10 data points shown in Fig. 6(left). Each time, we run the three batch approximation methods on a batch of three randomly sampled points. We repeat this 1000 times and show their empirical evaluations in Fig. 6(right). Compared with the unbiased MWS estimator, MMS and IS are unbiased. Compared with MMS, the IS estimator has low empirical variance across 1000 repeats, which implies a more stable estimation.

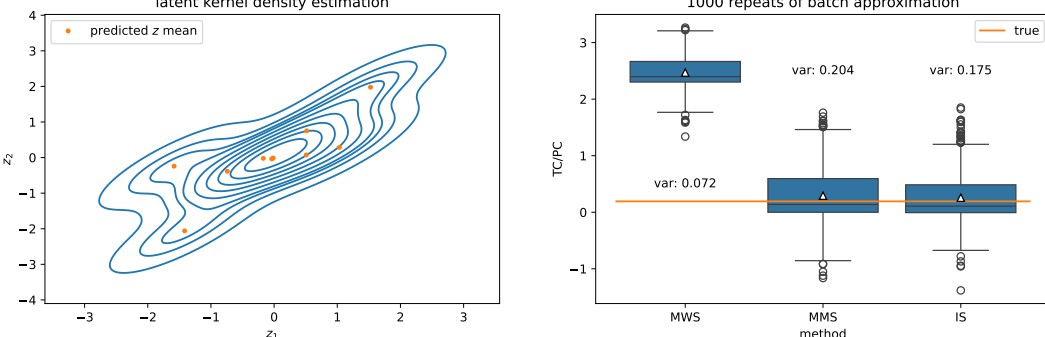

Figure 6: **Left**: Predicted mean of the latent $\boldsymbol{z} = (z_1, z_2)$ and its kernel density estimation. **Right**: 1000 repeats of batch approximations by the three methods, their empirical variance across the 1000 repeats.

## A.4 SUPPLEMENTARY RESULTS

### A.4.1 ABLATION

To analyze the choice of the penalty coefficient $\beta$ of PC term in Eq. (3), we vary $\beta$ in PDisVAE from 0.1 to 100 and plot the cross-validation results in Fig. 7. The PC and latent $R^2$ plots indicate that $\beta > 1$ is necessary for an accurate recovery and effective minimization of the PC. However, excessively large $\beta$ might negatively impact reconstruction, as shown in the reconstruction $R^2$ plot. Hence, we recommend $\beta \in (2, 10)$, which supports our choice of $\beta = 4$ in our experiments.

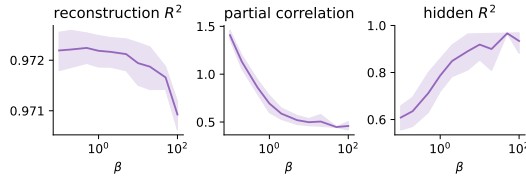

Figure 7: Three metrics w.r.t. the PC coefficient $\beta$ in PDisVAE.

### A.4.2 FLEXIBLY REDUCE TO THE FULLY INDEPENDENT CASE

**Dataset and experimental setup.** To validate that PDisVAE can get the same results as from a fully disentangled VAE when the latent is fully independent, we create a dataset consisting of $N = 2000$ points in $K = 3$ latent space $\boldsymbol{z}^{(n)} \in \mathbb{R}^3$, where the three latent components are independent with each other $z_1 \perp z_2 \perp z_3$. Their distributions are shown in Fig. 8(a) and Fig. 9. The observation $\boldsymbol{x}$ is linearly mapped from the latent $\boldsymbol{z}$ to a $D = 20$ dimensional space $\boldsymbol{x}^{(n)} \in \mathbb{R}^{20}$, and then Gaussian noise $\epsilon_d^{(n)} \overset{i.i.d.}{\sim} \mathcal{N}\left(0, 0.5^2\right)$ are added. Although we only have $K = 3$ true latent components, we still learn $K = 6$ components to compare their flexibility when the true number of latent components is unknown. The experimental setup is the same as the previous one.

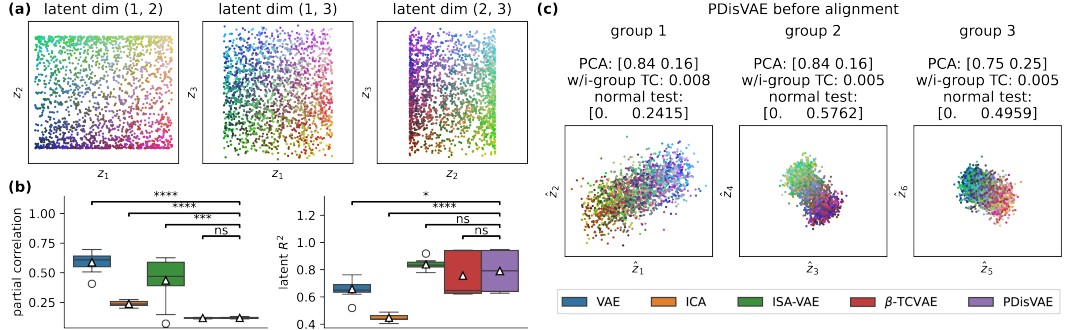

Figure 8: **(a)**: The true latent $\boldsymbol{z} \in \mathbb{R}^3$ coded by RGB = $z_1 z_2 z_3$, where three components are $z_1 \perp z_2 \perp z_3$. **(b)**: The PC of the estimated latent and the latent $R^2$ after alignment to the true latent in (a). The $t$-test between PDisVAE and others shows that PDisVAE is similar to ISA-VAE and $\beta$-TCVAE (ns: $p > 0.5$, *: $p \leqslant 0.05$, ****: $p \leqslant 0.0001$). **(c)**: The estimated latent of PDisVAE before aligning to the true latent shown in (a). The arrow in each plot shows the embedded true latent direction.

**Results.** The PC box plot and latent $R^2$ plot in Fig. 8(b) show that ISA-VAE, $\beta$-TCVAE, and PDisVAE achieve the lowest partial correlation and the highest latent $R^2$ on this fully disentangled dataset, which implies that PDisVAE automatically reduces to fully independent result if the group rank is deficient. In general, the actual group rank can be detected by PDisVAE and if the true group rank is less than the specified group dimensionality, dummy estimated latents will be complemented in the corresponding group. Due to the strong requirement in ICA that tries to find logcosh-independent components, but only three exist, ICA is not able to correctly identify three and find three dummy dimensions. This means logcosh might be too strong to allow the existence of dummy variables, which

could be harmful when we do not know the true number of latent components. Fig. 9 also visually shows that ISA-VAE, $\beta$-TCVAE, and PDisVAE accurately estimate the three latent distributions the best, which is consistent with the latent $R^2$ plot in Fig. 8(b).

To identify the three dummy latent dimensions complementing the three groups respectively through an unsupervised approach, we plot the PDisVAE result before alignment in Fig. 8(c). First, within-group TCs are all very small. Since "independence is non-Gaussian", we can find a direction within each group that yields $p > 0.05$, which accepts the null hypothesis of the normal test that a Gaussian noise dummy dimension exists. The arrows in Fig. 8(c) also visually indicate the embedded true latent direction.

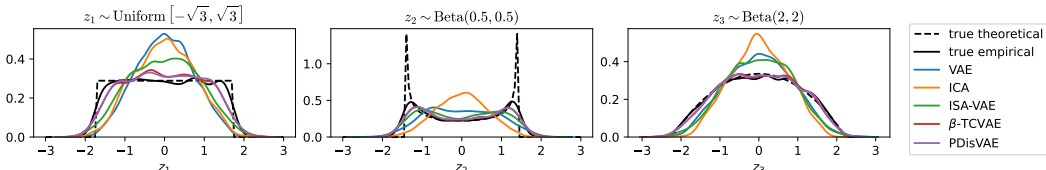

Figure 9: Estimated and true latent distribution after alignment to the true latent shown in Fig. 8(a).

### A.4.3 SYNTHETIC APPLICATION: PARTIAL DSPRITES

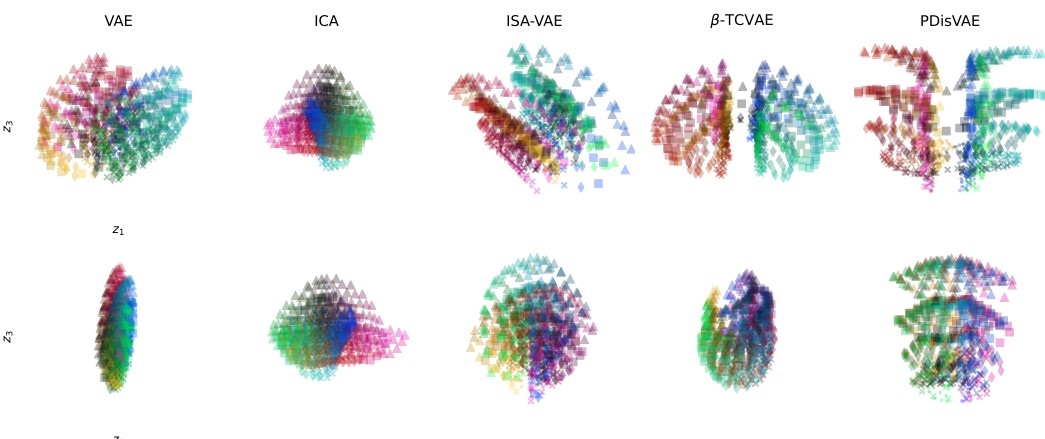

Figure 10: The latent plot after alignment in latent space $(z_1, z_3)$ and $(z_2, z_3)$ for different methods. The color representation for location is the same as the color representation in Fig. 4(a), and the marker of the point in the latent plots represents the size of the square in the observation images.

Table 4: The PC, latent $R^2$, latent MSS, and adapted mutual information gap (MIG) evaluated for different methods on the dsprites dataset.

|  | PC ↓ | $R^2$ ↑ | MSE ↓ | MIG ↑ |
|---|---|---|---|---|
| VAE | 1.01 (0.02) | 0.22 (0.04) | 0.29 (0.02) | 0.15 (0.01) |
| ICA | 1.76 (0.07) | 0.22 (0.06) | 0.28 (0.03) | 0.14 (0.09) |
| ISA-VAE | **0.70 (0.01)** | 0.23 (0.02) | 0.33 (0.01) | 0.24 (0.08) |
| $\beta$-TCVAE | 0.91 (0.10) | 0.33 (0.06) | **0.24 (0.04)** | 0.36 (0.13) |
| PDisVAE | **0.68 (0.04)** | **0.54 (0.08)** | **0.23 (0.04)** | **0.49 (0.07)** |

## A.4.4 REAL-WORLD APPLICATIONS

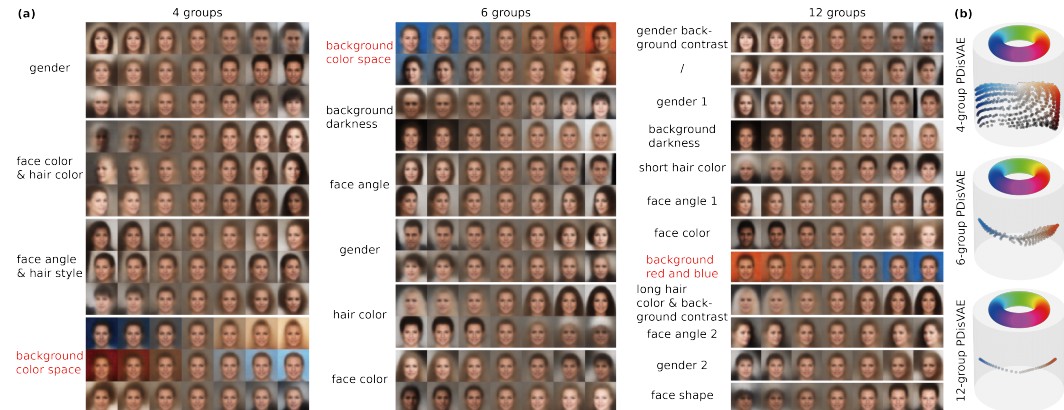

Figure 11: **(a)**: Reconstructed images are shown by varying one of the $K = 12$ latent dimensions from PDisVAE applied to the CelebA dataset, with different numbers of groups $G \in \{4, 6, 12\}$. Each row corresponds to varying one latent component (dimension) while fixing all others to 0s. **(b)** The spanned color space by the red-annotated color group in the $\{4, 6, 12\}$-group PDisVAE.

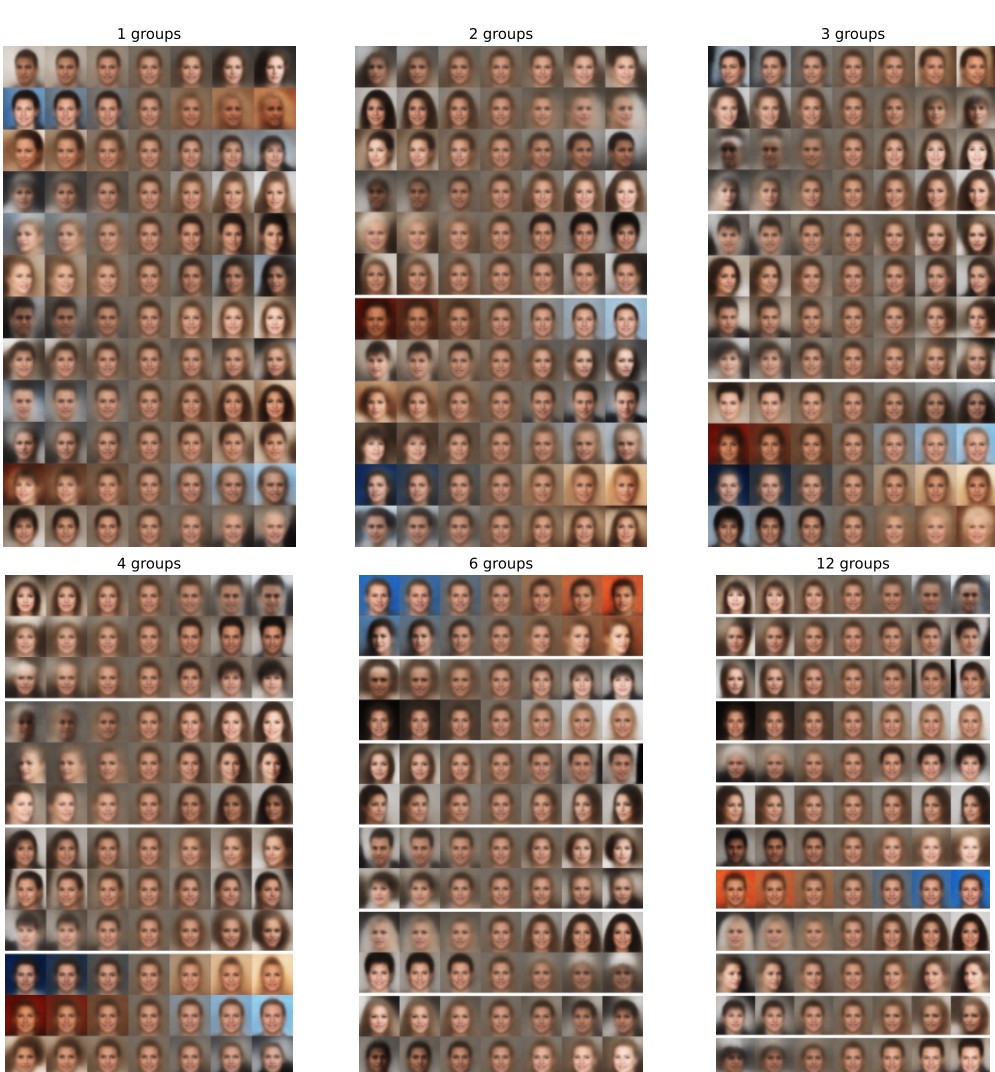

Figure 12: The reconstructed images by varying one of the $K = 12$ disentangled latent from applying PDisVAE to the CelebA dataset with the different number of groups $G \in \{1, 2, 3, 4, 6, 12\}$. When $G = 1$, PDisVAE becomes the standard VAE; when $G = K = 12$, PDisVAE becomes the fully entangled VAE (e.g., $\beta$-TCVAE or FactorVAE). In each plot, each row is by varying one latent component (latent dimension) while fixing all others to 0s.

## A.5 MORE DISCUSSIONS ABOUT INTERPRETING SEMANTIC VS. STATISTICAL INDEPENDENCE IN PRACTICAL APPLICATIONS

In a lot of practical applications, we need to differentiate two concepts: semantic meaning vs. statistically independent group. It is possible that an independent group contains more than one semantic meaning. In the CelabA dataset, for example, it is likely that females have more warm backgrounds and males have more cold backgrounds. In this case, the background warm/cold is entangled with gender. In this case, we cannot separate these two semantic meanings since they are statistically dependent/entangled.

In our example of background color, especially Fig. 11(b), we interpret a group as background color based on our human understanding. However, we cannot rigorously prove that the background color is totally independent of the tiny facial feature changes. This is actually an important point we want to stress in this paper, like in Sec. 1 paragraph 2, Fig. 4(a), and Fig. 11(b). We can summarize the following four possibilities:
• one semantic meaning corresponds to one latent component (fully independent);
• one semantic meaning corresponds to several entangled latent components (a latent group);
• several semantic meanings correspond to one latent component (semantic meanings are entangled and encoded by one latent component);
• several semantic meanings correspond to several latent components (semantic meanings are entangled and encoded by several latent components).
This is the key reason we generalize fully disentangled VAE to partially disentangled VAE (PDisVAE) since PDisVAE considers all these possibilities that exist in nearly all real-world datasets (maybe with the probability of 1). We view this as our paper's key take-home message that we really need to jump out of the stereotype that one latent component should correspond to one semantic meaning.

For example, in the partial dsprites (pdsprites) dataset shown in Fig. 4(a), although we humans think $x$ location and $y$ location are two separable semantic meanings, they are statistically dependent/entangled with each other, so we cannot separate them but put them in one group, and that is why fully disentangled VAEs (e.g., $\beta$-TCVAE) fails with this dataset (Fig. 4(b)). We can think $x$ and $y$ as two semantic meanings or say $(x, y)$ "location" is one semantic meaning, but the ground truth is that $x$ location and $y$ location are entangled, not statistically separable, and hence should be encoded by a latent group of at least rank-2.

A similar reason also holds for the color distribution we plot in Fig. 11(b). If we use a fully disentangled VAE, we can only interpret that the background color (from red to blue, a curve in HSV space) is encoded by one latent component, but that might not be the fact. We do show in Fig. 9(b) that with more latent components entangled with each other as a group, the background color semantic meaning can be expressed more fully (a 2D manifold or a restricted 3D region that is not evenly distributed).

Therefore, no one can promise an absolutely perfect correspondence between semantic meaning(s) and a latent component/group. All researchers can do is validate the correctness of their method on synthetic datasets, as we do in Sec. 4.1, and get more interpretable (but cannot promise perfect correspondence) disentanglement results on real-world datasets. Generally speaking, it is nearly impossible for all kinds of disentangling methods to find pure correspondence between a latent component/group and one semantic meaning on real-world datasets. At least there are some noises, including other semantic meanings of tiny magnitude. This kind of result should be acceptable in the field of representation learning (disentanglement), especially on real-world datasets where there is no true latent. Otherwise, any interpretation from any method could have small flaws (that can even come from random seeds or the floating point precision of the training device).

## A.6 USE OF LARGE LANGUAGE MODELS

We used large language models (LLMs) solely to aid in writing polish and minor language improvements (e.g., fixing grammar issues, rewriting sentences in a more formal style. They were not used for scientific exploration, conceptualization, experimental design, analysis, or conclusions.

