# OpenReview forum: "A Revisit of Total Correlation in Disentangled Variational Auto-Encoder with Partial Disentanglement"
_ICLR.cc/2026/Conference — Submitted to ICLR 2026_

### Official Review · Reviewer_tj5C · 2025-10-23

**Soundness:** 2
**Presentation:** 3
**Contribution:** 3
**Rating:** 4
**Confidence:** 4

**Summary:**

This paper introduces an approach that tries to relax the full independence assumption in traditional disentangled VAEs to accommodate group-wise independence - a more realistic scenario where semantically meaningful factors my be inherently entangled within groups but independent across groups. The key innovation is the generalization of the total correlation (TC) penalty used in fully disentangled VAEs to a partial correlation (PC) penalty, which explicitly enforces independence between groups while flexibly permitting within-groups entanglement, allowing in principle the within-group disentanglement rank to vary.

The authors present an argument that attempting to disentangle multiple semantically meaningful factors may be a difficult endeavor, and that jointly encoding such entangled factors independently from another group of entangled semantically meaningful factors constitutes a more principle approach.

**Strengths:**

The work tackles an important representation learning problem by allowing statistically independent factors to be encoded as vector-valued rather than scalar variables, thereby enabling more expressive and efficient latent representations.

The theoretical argument supporting importance sampling, as compared to mini-batch weighted sampling and mini-batch stratified sampling, appears technically sound and adequately justified.

The proposed method may help to identify the underlying relationship between the entangled factors encoded in a single group.

**Weaknesses:**

Even when the objective is to encode statistically independent groups of factors as vectors instead of disentangling on the basis of semantics, additional inductive biases are still required, since nonlinear ICA is fundamentally unidentifiable under the independence assumption alone.

The synthetic datasets employed for validation are limited to linear mappings from the representation space to the data space, which substantially constrains the generality and scope of the validation. This is evident from the weaker results on the nonlinear mapping, partial dsprites dataset.

While the method aims to encode statistically independent factor groups, the learned representations seem to violate the group-wise independence (especially for the partial dsprites dataset) by capturing information from multiple factor groups simultaneously, substantially reducing interpretability.

The partial dsprites and the synthetic validation on the "flexibly reduce to the fully independent case" is not convincing (ficure 4c group 2) and it seems that the rank does not reduce to the true number of components, which greatly reduces the interpretability of the method.

It is very hard to interpret the results for Celeb-A and to know which groups of semantic meanings are encoded by which latent variables (and vice versa).

A logcosh-priored VAE does not correspond to an equivalent nonlinear ICA problem since independence is insufficient for source recovery under nonlinear mixing.

**Questions:**

What is the R2 alignment matrix for the partial dsprites dataset?

Provide ablation studies with nonlinear mixing datasets

What are the values of the disentanglement metrics, specifically the ones which measure independence (DCI, MIG, modularity, ...) on the partial dsprites dataset (appropriately adjusted to account for group independence)

Clarify the calculation of PC on the true latent (ground truth?)

---

> ### Author Response · Authors · 2025-11-21
>
> Dear Reviewer tj5C,
>
> Thank you very much for your time and valuable comments on our paper. We highly appreciate your recognition of the strengths of our paper. Hopefully, the following responses could resolve most of your concerns and answer your questions.
>
> ## Weaknesses
> ### W1 and W6
> We acknowledge that "nonlinear ICA is fundamentally unidentifiable under the independence assumption alone". We apologize if we did not make this clear, but we never claim we want to achieve identifiability. The main focused contribution of this paper is to generalize the fully disentangled VAE (i.e., $\beta$-TCVAE and FactorVAE) to a partially disentangled VAE, which is more flexible and a more general assumption. This line of research does not conflict with "adding supervision to achieve identifiability", but this is out of the scope of this paper. What we want to resolve in this paper is a generally better way of assuming and discovering independent latent structure unsupervisedly, as is clearly mentioned in the first sentence of the abstract. Certainly, this is not saying identifiability and supervision are not important, and does not hinder readers from adding supervision to the latent for inductive bias.
>
> ### W2
> We kindly do not think this is a weakness, but a proper way of conducting experiments. A simple linear synthetic experiment is necessary for readers and us to validate and understand the behavior details of a newly proposed method, followed by a complicated non-linear synthetic experiment, and then some real-world datasets. We should not compare performances between datasets or the generative models (VAE neural network architectures), but compare performances between different methods. On complicated VAE models and challenging datasets, all methods face the same challenge and perform worse, which is under expectation.
>
> ### W3
> We apologize that we did not get this question clearly. Fig. 4 clearly shows that our PDisVAE gets significantly better results (both metrics and visualization interpretations) than other baseline models. Meanwhile, it is acceptable that a machine learning model does not achieve their goals 100%. Even if we want to achieve independence between groups, we cannot promise that the final evaluation of the partial correlation between groups is precisely 0, at least due to the numerical issue. Even for all other baseline disentangled VAE methods, their post-hoc independence/disentanglement evaluation cannot be 0 perfectly.
>
> ### W4
> Please kindly remind that "flexibly reduce to the fully independent case" means setting $G=K$, PDisVAE becomes a fully disentangled VAE. I.e., setting $G=K$, the target function of PDisVAE is exactly the same as $\beta$-TCVAE and FactorVAE. In other words, a fully disentangled VAE is a special case of PDisVAE when each latent dimension is a group. This is a mathematical unification and is unrelated to any experimental result.
>
> ### W5
> We apologize that we did not get this concern clearly. First, Fig. 11(a) and (b) clearly annotate the interpretation of the latent groups, which is based on human interpretation of visualization, the same as other disentangled VAE papers (e.g., [Fig. 1 in $\beta$-TCVAE](https://arxiv.org/pdf/1802.04942)).  Second, in Appendix A.5, we have a comprehensive discussion regarding interpreting semantic vs. statistical independence in practical applications. These discussions are focused on the practical usage of disentangled VAE in general, not only on our PDisVAE. Therefore, these discussions and results provide insights into the field of unsupervised representation learning, rather than weakening the effects of our proposed method.

---

> ### Author Response · Authors · 2025-11-21
>
> ## Questions
> ### Q1
> | VAE | ICA | ISA-VAE | $\beta$-TCVAE | PDisVAE |
> | - | - | - | - | - |
> | $\begin{bmatrix} 0.2752 & 0.0825 \\\\ 0.0784 & 0.1727 \end{bmatrix}$ | $\begin{bmatrix} 0.0124 & 0.3587 \\\\ 0.0792 & 0.0019 \end{bmatrix}$ | $\begin{bmatrix} 0.0650 & 0.4859 \\\\ 0.2059 & 0.0011 \end{bmatrix}$ | $\begin{bmatrix} 0.5226 & 0.0475 \\\\ 0.0326 & 0.1445 \end{bmatrix}$ | $\begin{bmatrix} 0.6862 & 0.0001 \\\\ 0.0012 & 0.3946 \end{bmatrix}$ |
>
> ### Q2
> Please kindly note that our comprehensive experiments on pdSprites and CelebA are all nonlinear mixing datasets. The generative model (decoder) and the encoder are all non-simple neural networks (Burgess and ImageNet).
>
> ### Q3
> As we explained in line 256, no matter on synthetic datasets (with known ground truth) or on real-world datasets (without ground truth), we can always use PC-on-test-set to evaluate the group-wise independence. When the ground truth exists on synthetic datasets, we can also evaluate $R^2$ between the estimated latent groups with the ground truth groups. For the pdSprites dataset, both $R^2$ and PC values on the test set are displayed in Fig.3 (c). Other supplementary metrics are redirected to Tab. 4 in Appendix A.4.3.
>
> ### Q4
> Evaluating partial correlation (PC) on ground truth latent follows exactly the same way as on estimated latent. Estimating PC does not differentiate whether the latent is estimated from a model or from the ground truth. Specifically, with latent data points $\\{\boldsymbol z^{(n)}\\}_{n=1}^N$, we can use Eq. (4) to evaluate all terms of the form $q(z)$ and insert them into the PC term in Eq. (3). The details of this calculation is also provided in our code (in the supplementary material).

---

### Official Review · Reviewer_VLKy · 2025-10-31

**Soundness:** 3
**Presentation:** 2
**Contribution:** 2
**Rating:** 4
**Confidence:** 4

**Summary:**

This paper proposes PDisVAE, a framework for partial disentanglement in Variational Autoencoders (VAEs). The core idea is to generalize the Total Correlation (TC) penalty used in fully disentangled VAEs to a Partial Correlation (PC) penalty, which enforces independence betweengroups of latent variables while allowing for entanglement withineach group. The authors provide a theoretical analysis of batch approximation methods, arguing for the superiority of Importance Sampling (IS), and validate their method on synthetic and real-world datasets.

**Strengths:**

1. The paper correctly identifies that enforcing full independence among all latent factors can be an overly strong and unrealistic inductive bias. Exploring partial, group-wise disentanglement is a meaningful direction for the field.
2. The claim that the proposed framework can reduce to a standard VAE (G=1) or a fully disentangled VAE (G=K) is a theoretically appealing property, suggesting a generalized approach.

**Weaknesses:**

1. The introduction spends too much time on well-known background and could be restructured to more sharply highlight the gap in the literature (the lack of flexible, penalty-based methods for partial disentanglement) and the paper's specific contributions.
2. The explanation of the PC term is abrupt. A more intuitive explanation of how the PC penalty operates during training would be helpful. How does it practically encourage the formation of groups?
3. The comparison to ISA-VAE is a critical point but is not fully leveraged. The paper correctly states that ISA-VAE uses a rigid prior (Lp-nested), while PDisVAE offers flexibility. However, the experiments, while showing PDisVAE's superiority, do not deeply analyze whyISA-VAE performs poorly on the synthetic tasks. Is it solely the prior mismatch? A more detailed discussion contrasting the prior-based vs. penalty-based approaches would strengthen the paper's contribution.
4. A significant practical challenge is choosing the number of groups G. The paper mentions this as a limitation in the conclusion but defers it to future work. While it's a valid and difficult problem, providing some practical guidance or even a simple heuristic (e.g., based on the PC value on a validation set) would greatly increase the practical utility of the method.

**Questions:**

1. How does the computational cost (training time, memory) of PDisVAE scale with the number of groups G and the chosen batch approximation method compared to a standard β-TCVAE?
2. Could the performance gap between PDisVAE and ISA-VAE on your synthetic datasets be closed by using a more flexible family of priors within the ISA-VAE framework, or is the penalty-based approach fundamentally more adaptable?
3. In a real-world scenario with no ground truth, how should a practitioner choose the number of groups (G) and the per-group dimensionality? Did you perform any sensitivity analysis on these hyperparameters for the real-world experiments?
4. Can you demonstrate the utility of the partially disentangled representations learned by PDisVAE on a downstream task? For example, on CelebA, could you show that your method allows for more precise control and better generalization in attribute manipulation compared to a fully disentangled VAE?

---

> ### Author Response · Authors · 2025-11-21
>
> Dear Reviewer VLKy,
>
> Thank you very much for your time and valuable comments on our paper. We highly appreciate your recognition of the strengths of our paper. Hopefully, the following responses could resolve most of your concerns and answer your questions.
>
> ### W1
> We apologize that the introduction might be too long and detailed for some well-known background knowledge. We will adjust the introduction accordingly to stress more on our own contributions.
>
> ### W2
> We kindly believe that why PC encourages group-wise independence is well-explained in line 160 and Fig. 1. Since TC encourages dim-wise independence, generalizing TC to PC for group-wise independence is intuitive and follows the same mathematical principle that we want to minimize the KL divergence between the joint $q(\boldsymbol z)$ and the product of group marginals $q(\boldsymbol z_g),\ g\in\\{1,\dots,G\\}$.
>
> ### W3 and Q2
> The fact that finding a group-wise independent prior distribution is theoretically hard. The Lp-nested prior distribution is the only existing group-wise independent prior family (i.e., only the ISA-VEA paper proposes that), up to the author's knowledge. The purpose of this paper is to point out the limitations of the rigid Lp-nested prior. Through our experiments, we are confident to answer that the penalty-based approach is fundamentally more adaptable, because the latent distribution is based on latent data (i.e., kernel density estimation), which adapts to arbitrary latent space structure. As pointed out in line 315, although ISA-VAE is designed to find group-wise independent latent, its performance is not ideal when facing data generated from group-wise independent ground truth latent in practice, due to the predefined group-wise independent prior in ISA-VAE differing substantially from the true underlying latent groups. And in practice, we do not know other group-wise independent prior distribution family other than the Lp-nested prior.
>
> ### W4 and Q3
> We would like to kindly remind that we not only mention "choosing $G$ and $H$" is a potential challenge, but also stressed in line 193 and line 483 that setting a large enough $G$ and $H$ is theoretically feasible, because redundancy will be detected during the training process, and we can diagnose those redundancies in a post-hoc way. If a group is totally redundant, it will be more like Gaussian noise, which can be checked by the statistical normal test. If $H > H_{\mathrm{true}}$, it will automatically detect the true effective group rank with the remaining $H - H_{\mathrm{true}}$ dimensions as dummy variables. This flexibility
> and effectiveness will be validated through experiments (Fig. 2, Fig. 8 in Appendix A 4.2, and Fig. 4 group 2 size, which is a real-world scenario).
>
> ### Q1
> Thanks for this point. For a batch with $M$ data points, $G$ groups in $K$ dimensional latent space, the complexity of evaluating the product of marginal groups $q(\boldsymbol z_g),\ g\in\\{1,\dots,G\\}$ is $\mathcal O(G \times M^2 \times K/G) = \mathcal O(KM^2)$. Specifically, $G$ numbers of $q(\boldsymbol z_g)$. Each one has $M$ points, and each point has a sample, so we are evaluating $M$ samples on $M$ Gaussian kernels. Each Gaussian kernel evaluation is a density of $K/G$ dimensions. The final outcome shows that it is the same as the special case of TC in $\beta$-TCVAE when $G = K$. In our code, the only difference is the order between `sum` and `logsumexp` of the Gaussian KDE. So, the complexity is exactly the same as $\beta$-TCVAE.
>
> ### Q4
> The motivation for conducting the CelebA experiment is explained in the second paragraph of the introduction. As explained in Fig. 11(a) and (b), the background color encoded by a 1-dimensional latent is not representative enough. Reading from the CSV color space in Fig.11(b), we can see that three latent components as an entangled group (cannot be separated further) are more expressive for the semantic meaning of background color. We would like to remind the author that a more detailed discussion on the downstream application, especially for the pdSprites and CelebA datasets, is in Appendix A.5.

---

### Official Review · Reviewer_3jNW · 2025-11-01

**Soundness:** 2
**Presentation:** 2
**Contribution:** 2
**Rating:** 4
**Confidence:** 3

**Summary:**

PDisVAE replaces the TC penalty with a PC term to enforce group-wise independence while allowing within-group entanglement.

**Strengths:**

1. The method proposed in this paper can degenerate into the standard VAE and fully disentangled VAEs.
2. The paper not only presents theoretical results but also experimental results.

**Weaknesses:**

1. From Eq 3 the paper defines the PC term. However, the paper does not explain how this quantity is practically estimated. It jumps directly to Eq 4, which proposes a batch approximation to the density q(z), and then states Theorem 3.1 claiming that IS is an unbiased estimator for q(z) with lower variance than MWS/MSS. This is presented as a key contribution. But there might be two issues here:

(a) Theorem 3.1 is about the density level (estimating q(z)). It does not directly imply anything about the final objective, the PC term, which is a function of log-densities. Even if $q̂(z)$ is unbiased for q(z), the estimator $log q̂(z)$ is generally not unbiased for log q(z) because of Jensen’s inequality (equality holds only in degenerate cases). Hence, unbiasedness does not transfer to the PC term.
For variance, one might hope that smaller $Var[q̂(z)]$ imply a smaller variance of $\hat{PC}$. But this derivation is not trivial. It need further analysis. Since the paper does not provide this analysis, Theorem 3.1 gives limited support for statistical claims about the PC estimator (core proposal of the paper).

(b) Readers can not see any explicit formula or introductions for the estimator used for the PC term. For estimating PC term, there might be two ways: one is like FactorVAE, directly estimating density-ratio through additional network. One is first estimate qz then take logs and .... I guess authors chose the latter one. Therefore it would be better to show a comprison (theoretical or empirical) between these two ways focusing on their estimation performance with respect to the PC term rather than qz.



2. For Fig 3a, it seems it is a synthetic experiment, thus we expect PC term to be zero. Could you discuss why the result is mean:0.332 (std: 0.006), which is far away from 0?

3. line 437, 'RMSE= 0.47', but in fig 5 RMSE are around 0.046

4. A minor comment:This paper replaces the TC term with a new term to enforce group-wise independence, and names this term "partial correlation". However, this choice of terminology is problematic, because "partial correlation" is already a well-established technical term in statistics and machine learning. Maybe use another word if possible, or state their difference.

**Questions:**

See weaknesses.

---

> ### Author Response · Authors · 2025-11-21
>
> Dear Reviewer 3jNW,
>
> Thank you very much for your time and valuable comments on our paper. We highly appreciate your recognition of the strengths of our paper. Hopefully, the following responses could resolve most of your concerns and answer your questions.
>
> ## Weaknesses
> ### W1
> For (a), we understand that due to Jensen's inequality, unbias in original space does not imply unbias in log space. However, what we want to contribute here is to show that IS can make some components in the estimation more stable than other methods. We understand that proving the final $\hat{PC}$ will be more stable with IS is rigorously hard and non-trivial, but showing a commonly used term $q(z)$ in $\hat{PC}$ can be more stable with IS, rather than others could provide readers a general guidance. Besides, we have a practical result in Figure 6 showing that IS is better than the others, at least for estimating PC/TC terms in disentangled VAE.
>
> For (b), yes, we choose the latter one, since we follow the same terminology as $\beta$-TCVAE using mini-batch estimation for $q(z)$. The reason we did not choose the way from FactorVAE is that we believe the comparison has been made in the [$\beta$-TCVAE paper](https://arxiv.org/pdf/1802.04942) and some follow-up [comparison works](https://arxiv.org/pdf/1911.11791). Existing results from these works have shown that the mini-batch estimation is
> * easy to train and converge,
> * and computationally lighter, since no auxiliary discriminator network is needed.
>
> Therefore, we assume these supportive preliminary results have provided clear guidance for adopting this easy but good batch method for estimating PC.
>
> ### 2
> As we mentioned in line 300, the PC is evaluated numerically. Theoretically, it should be zero. Hence, when we see a PC value evaluated from a learned latent, we need to have a sense of the PC magnitude under such a dataset. Therefore, using the same Monte Carlo samples to evaluate the PC on the ground truth could provide us a baseline, i.e., 0.332 ± 0.006, indicating that this magnitude implies close to zero for this particular dataset. Then, when evaluating PC from the learned latent from different methods, we know that our PDisVAE's PC = 0.6, which is indeed a more group-wise independent latent structure than plain VAE's PC = 1.5.
>
> ### 3
> We apologize for this typo. In the main text, it should be 0.047.
>
> ### 4
> Thanks for this suggestion. We think "partial correlation" is a generalization of "total correlation". So, we would like to clarify its difference w.r.t. the existing "partial correlation" in terms of linear relationships in machine learning.

---

### Meta-Review · Area_Chair_ccZy · 2026-01-06

**Summary:**

Reviewers raised concerns primarily related to clarity, methodological rigor, and empirical support. Several reviewers noted that the paper lacks sufficient practical detail on how the proposed quantity is estimated, including the absence of concrete algorithmic descriptions or pseudocode, which limits reproducibility and practical understanding (3jNW).
Other reviewers questioned whether the paper provides a convincing explanation for why the proposed quantity encourages disentanglement, and whether the role and selection of key hyper-parameters (e.g., the number of groups) are sufficiently justified with quantitative evidence (VLKy).
The authors clarified that the scope of the work is disentanglement and explained the use of linear synthetic settings as verification, which addresses part of the experimental-design concerns.

 Overall, however, the paper would benefit from revisions that (1) clearly specify the estimation procedure and (2) provide more thorough analysis and quantitative support for how the proposed quantity improves disentanglement.

**Reviewer Concerns:**

### Addressed

1. **Scope clarification:**
   The authors clearly stated that the paper does not focus on identifiability, addressing concerns about the intended scope of the work.

2. **Linear synthetic experiments:**
   The authors explained that the linear synthetic experiments are included solely for verification purposes, not as a claim about general applicability.

### Outstanding

1. **Unclear estimation procedure:**
   The paper still lacks a clear and detailed description of how the proposed quantity is estimated in practice, including algorithmic steps or pseudocode.

2. **Insufficient analysis of the proposed quantity:**
   The paper does not provide a thorough analysis explaining why the proposed quantity improves disentanglement, nor does it offer sufficient quantitative evidence demonstrating flexibility with respect to key hyper-parameters.

**Reviewer Scores:**

Reviewer **3jNW** raised concerns about the lack of detailed practical estimation of the proposed quantity. While the authors provided a brief explanation, they did not include concrete algorithmic details such as pseudocode or a clear estimation procedure. As a result, this concern is not fully addressed, and the reviewer is likely to remain negative.

Reviewer **VLKy** expressed concerns that the paper does not sufficiently explain how the proposed quantity encourages disentanglement, and requested clearer justification for the choice and impact of the hyper-parameter controlling the number of groups. Although the authors briefly explained the intuition and pointed to Appendix 4.2 for hyper-parameter studies, there are no quantitative comparisons supporting the claimed flexibility. Consequently, this reviewer is also likely to remain negative.

Reviewer **tj5C** questioned the non-identifiability of nonlinear ICA without additional conditions and the reliance on linear settings in synthetic experiments. The authors clarified that the scope of the paper is disentanglement rather than identifiability, and explained that the synthetic experiments are meant for verification. They also pointed out the inclusion of nonlinear experiments. This response is likely sufficient for the reviewer to maintain or increase their score.

---

### Decision · Program_Chairs · 2026-01-26

Reject